# Ayurgenomics and Modern Medicine

**DOI:** 10.3390/medicina56120661

**Published:** 2020-11-30

**Authors:** Robert Keith Wallace

**Affiliations:** Department of Physiology and Health, Maharishi International University, Fairfield, IA 52556, USA; kwallace@miu.edu

**Keywords:** Ayurgenomics, Ayurveda, genomics, P4 medicine, diet, lifestyle, disease, prevention, personalized medicine

## Abstract

Within the disciplines of modern medicine, P4 medicine is emerging as a new field which focuses on the whole patient. The development of Ayurgenomics could greatly enrich P4 medicine by providing a clear theoretical understanding of the whole patient and a practical application of ancient and modern preventative and therapeutic practices to improve mental and physical health. One of the most difficult challenges today is understanding the ancient concepts of Ayurveda in terms of modern science. To date, a number of researchers have attempted this task, of which one of the most successful outcomes is the creation of the new field of Ayurgenomics. Ayurgenomics integrates concepts in Ayurveda, such as Prakriti, with modern genetics research. It correlates the combination of three doshas, Vata, Pitta and Kapha, with the expression of specific genes and physiological characteristics. It also helps to interpret Ayurveda as an ancient science of epigenetics which assesses the current state of the doshas, and uses specific personalized diet and lifestyle recommendations to improve a patient’s health. This review provides a current update of this emerging field.

## 1. Introduction

Ayurveda is the ancient system of traditional medicine in India. The term Ayurveda comes from two Sanskrit words, “ayus”, meaning life or lifespan, and “Veda”, meaning knowledge or science. Ayurveda may be translated as “the science of life”, or more specifically, “the science of lifespan”. Ayurveda was originally an oral tradition of natural health, and it was much later that this knowledge was written down in books [1]. Some of the knowledge became fragmented and lost due to many years of foreign rule in India.

The Tridosha theory of Ayurveda explains that there are three fundamental principles or forces, called doshas, which govern the physiology of each individual. Vata is the dosha involved in transportation in the body; from the transportation of molecules to the transportation of nervous impulses. It arises from the elements of ether and air. Pitta is the dosha that governs the process of digestion, as well as all metabolic pathways inside each cell. It is formed from fire and water. Kapha is the dosha that governs structure and cohesion in the body. It is an expression of earth and water. Each individual is born with a particular combination of these three doshas; this is called Prakriti. There are seven basic types of Prakriti: Vata; Pitta; Kapha; Vata/Pitta; Pitta/Kapha; Vata/Kapha; and Vata/Pitta/Kapha.

Is there a scientific explanation for Prakriti? The best description so far has come from the new field of Ayurgenomics, which attempts to describe Prakriti types in terms of modern genetics and physiology.

## 2. Studies on the Genetic Basis of Prakriti

A number of studies have correlated Prakriti with specific genetic and physiological measures. One early study in 2005 evaluated 76 subjects both for their Prakriti and human leucocyte antigen (HLA) DRB1 types. The researchers observed a correlation between HLA type and Prakriti type, with a complete absence of the *HLA DRB1*02* allele in the Vata type and of *HLA DRB1*13* in the Kapha type. Furthermore, HLA DRB1*10 had higher allele frequency in the Kapha type compared to the Pitta and Vata types [2].

In 2008, a comprehensive study was done correlating biochemical and genome wide expression levels in subjects from the three main Prakriti groups. There were many distinct differences in the regulation of genes in each of the main Prakriti groups. In Pitta types, for example, there was an over-expression of genes in the immune response pathways. In Vata males, there was an over-expression of genes related to cell cycles, particularly in the regulation of cyclin-dependent protein kinase activity and the regulation of enzyme activity. In Kapha males, there was a down-regulation of genes involved in fibrinolysis and an up-regulation of genes involved in ATP and cofactor biosynthesis. In addition, the researchers observed certain physiological differences such as Kapha types having higher levels of triglycerides, total cholesterol, high low-density lipoprotein (LDL) and low high-DL (HDL), compared to Pitta and Vata types. In Pitta types, they found hemoglobin and the red blood count were higher, whereas serum prolactin was higher in Vata types [3].

A study in 2010 studied high-altitude adaptation and common variations rs479200 (C/T) and rs480902 (T/C) in the EGLN1 gene. They found that the TT genotype of rs479200 was more frequent in Kapha types, and was correlated with a higher expression of EGLN1. In contrast, it was present at a significantly lower frequency in Pitta, and nearly absent in natives of high altitude. One of their interpretation is that Pitta types are more protected at high altitudes (see Table 1) [4].

Furthermore, a 2010 study showed that within Kapha types, there was a down-regulation of CYP2C19 genotypes, a family of genes that is involved in the detoxification and metabolism of certain drugs and an up-regulated in Pitta types [5]. In 2012, a paper appeared indicating that CD25 (activated B cells) and CD56 (natural killer cells) were higher in Kapha types [6].

A paper in 2012 studied rheumatoid arthritis patients and found that inflammatory genes were up-regulated in Vata types, while in Pitta and Kapha types there was an up-regulation of oxidative stress pathway genes [7]. In 2015, one of the most extensive and rigorous studies found that 52 SNPs (single nucleotide polymorphism) were significantly different in the three main types of Prakritis. Research also showed that the SNP (rs11208257) in PGM1 gene is correlated with energy production and is more homogenous and constant in Pitta than with Vata and Kapha types [8].

Another study done in 2015 identified DNA methylation signatures that distinguish the three major Prakriti types. The authors suggest that DNA methylation is probably coupled to chromatin regulation as a contributor to different Prakriti phenotypes, and that this research provides insight into the epigenetic mechanisms of the Ayurvedic personalized system of medicine [9].

One final paper in 2015 extended an earlier study, and found the combination of derived EGLN1 allele (HAPE associated) and ancestral VWF allele (thrombosis associated) was significantly high in the Kapha Prakriti group compared to the Pitta Prakriti group. They also showed a genetic link between EGLN1 and VWF, which could modify thrombosis/bleeding susceptibility and outcomes of hypoxia [10].

## 3. Studies on Physiology, Disease and Prakriti

A paper in 2003 showed that the Kapha Prakriti groups had a number of biochemical differences from Pittas or Vatas. They showed higher digoxin levels, increased glycoconjugate levels, increased free radical production and reduced scavenging, as well as increased tryptophan catabolites and reduced tyrosine catabolites [11]. In 2011, researchers found a significant decrease in the diastolic blood pressure immediately after isotonic exercise for five minutes in Vata-Kapha types, in contrast to Pitta-Kapha and Vata-Pitta types [12]. In another study, it was found that ADP-induced maximal platelet aggregation (MPA) was maximum among Vata-Pitta-type individuals [13].

Research reported in 2014 found a correlation between body mass index (BMI) and Prakriti type. It was reported that Kapha Prakriti subjects had a higher BMI than those with a Vata Prakriti [14]. A paper in 2015 found that Kapha types have higher parasympathetic activity and lower sympathetic activity in terms of cardiovascular reactivity, compared to Pitta or Vata types [15].

Several papers have also correlated disease conditions with Prakriti. A 2012 paper reported that Vata-Kapha types had significantly higher triglyceride, VLDL and LDL levels, as well as lower HDL cholesterol when compared with other constitution types. The researchers also found that Vata-Kapha Prakriti individuals had higher risk for diabetes mellitus, hypertension and dyslipidemia. In addition, Vata-Kapha types had higher levels of biochemical markers such as IL6, TNF alpha, hsCRP and HOMA IR. The Vata-Kapha and Kapha Prakriti groups were both correlated with higher levels of the inflammatory markers. The conclusion of the authors was that there is a high correlation of risk factors associated with the Vata-Kapha and Kapha Prakriti groups [16].

Another paper in 2012 looked at diabetic patients and studied the effects of walking (isotonic exercise). There was a significant decrease in systolic blood pressure in Vata-Pitta, Pitta-Kapha and Vata-Kapha types after walking. There was also a significant decrease in mean diastolic blood pressure in Vata-Pitta types [17]. Another paper found that the incidence of Parkinson’s disease was highest in patients with a Vata Prakriti [18]. In a 2015 study of patients with irritable bowel syndrome, it was found that most of the Vata type patients had developed IBS-C, or irritable bowel syndrome associated with constipation. The patients who were Pitta dominant developed IBS-D, or irritable bowel syndrome associated with diarrhea. The quality of life was found to be better in in Pitta-type patients [19]. Finally, several papers have focused on the connection between the microbiome composition and Prakriti, and found a significant correlation between specific bacteria groups and Prakriti [20,21,22].

## 4. Theoretical Papers

In addition to the research papers, there have been a number of theoretical and review papers. In 2008 and 2010, papers began promoting the idea of the connection between Ayurveda and genomics [23,24]. In 2011, a review and theoretical papers appeared in which the term Ayurgenomics was presented [25,26,27,28]. Since 2012, there have been many theoretical and review papers on Ayurgenomics from different angles [29,30,31,32,33,34,35,36,37,38,39,40,41]. In a few of these papers, the focus is also on related fields such as Pharmacogenetics and Ayurnutrigenomics [31,32]. In one 2016 paper, Ayurveda is described as an ancient science of epigenetics [42].

## 5. Modern Medicine and Ayurgenomics

Modern medicine uses a highly reductionist system to describe the fundamental basis of our physiology and health, using terms like genome, gene expression and epigenetics. Ayurveda uses an entirely different holistic system, which includes terms such as dosha and Prakriti.

Unfortunately, modern medicine has not recognized many of the useful preventative approaches of Ayurveda, due to a cognitive bias against folk or traditional medicine. Even though traditional systems of medicine are still widely used in many countries around the world, more research on their preventative and therapeutic treatment programs is necessary [43].

Up to now, the research has been primarily on specific herbal preparations, with the main outcome being an attempt to isolate one active ingredient which can then be used by a pharmaceutical company. Ayurgenomics offers a new bridge between traditional medicine and modern medicine by providing a rigorous scientific understanding of basic concepts, and at the same time incorporating the practical preventative approaches of Ayurveda into modern medicine.

Over the last ten years, a new field has arisen in modern medicine, which is known as P4 medicine. The four Ps are: predictive; preventive; personalized; and participatory [44,45,46,47]. This new system attempts to switch the emphasis from a disease-oriented system to a wellness-oriented system centered around the patient. It is closely related to other new fields such as Integrative Medicine, Functional Medicine, Lifestyle Medicine, Personalized Medicine and Preventive Medicine.

Ayurveda and other systems of traditional medicine have been patient-oriented and predictive, preventive, personalized and participatory for many thousands of years.

Long before the advent of epigenetic and other fields such as nutrigenetics, Ayurveda understood how diet and other lifestyle factors could affect our health. They recognized what modern medicine is only beginning to comprehend, that prevention is key to health. Improvements in diet, sleep, exercise and stress management are crucial for an effectively preventative system of medicine.

The development of Ayurgenomics, as we have said, helps give credibility to Ayurveda and other systems of traditional medicine by describing their ancient concepts in terms of modern science. New approaches in Ayurgenomics, which use big data analysis and machine learning, may greatly facilitate this whole process [36,37,39]. Of course, the therapeutic programs of these system will ultimately have to be tested through carefully controlled clinical trials.

Ayurveda and genomics can contribute to each other. Modern science can help Ayurveda as an evidence-based system of medicine, and Ayurveda can help modern medicine, particularly through its preventative approaches. This is especially true with P4 medicine, which is based on many of the same principles of Ayurveda. Having time-tested personalized preventative lifestyle recommendations would make it easy for each individual to participate in their own self-care (see Figure 1).

Ayurveda could also contribute to modern medicine in terms of the diagnosis of disease. Many believed that the enormous progress in the field of genomics would quickly bring a personalized system of medicine that could predict and prevent disease. Progress has been made, but researchers now realize that this may take longer than expected due to the highly complex nature of gene expression in the development of disease conditions. Ayurveda could help by stratifying individuals into broader categories using the Prakriti system of classification, along with modern genomics. Can we consider Prakriti as being our Ayurvedic genome? Can Ayurgenomics be used to diagnose and treat diseases?

These and many other questions will need to be answered by future research. We can only hope that a massive research effort is undertaken as soon as possible. The combination of Ayurveda and genomics promises to markedly improve many areas of healthcare throughout the world.

## Figures and Tables

**Figure 1 medicina-56-00661-f001:**
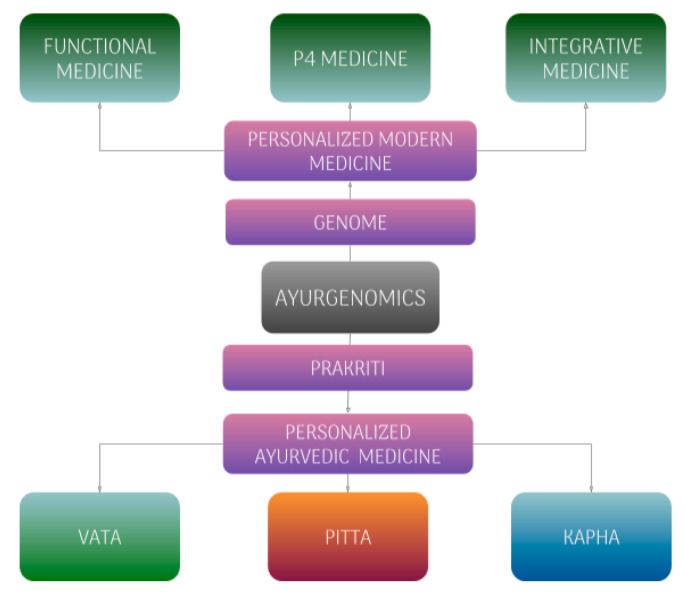
Ayurgenomics and its Relationship to Modern Medicine.

**Table 1 medicina-56-00661-t001:** Example of Prakriti and gene expression.

Prakriti Type	Gene Expression	Disease
Kapha/Vata	EGLN 1 higher	High Altitude Pulmonary Edema
Pitta	EGLN 1 lower	more adaptive for higher altitudes

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
