# Peer review of "Ayurgenomics and Modern Medicine"

_medicina, 2020, doi:10.3390/medicina56120661_

Round 1
Reviewer 1 Report
The Ayurgenomics article reads nicely and is easy to understand. I think we need to have some schematics like in this paper https://www.researchgate.net/publication/228439943_AYURGENOMICS_A_NEW_APPROACH_IN_PERSONALIZED_AND_PREVENTIVE_MEDICINE The graphic figure included is not sufficient.
Also, it would be nice to see a Table with some data on correlation between Dosha types and genes up-regulated, down-regulated or influenced. The problem of no standards available for Prakriti assessment should be highlighted. There is also inter-doctor variability in Prakriti assessment.
Note sure if these references need to be included: https://link.springer.com/article/10.1007/s40495-020-00244-3 https://journals.lww.com/progprevmed/Fulltext/2019/04000/Genomics_and_Big_Data_Analytics_in_Ayurvedic.1.aspx https://www.researchgate.net/profile/Niraj_Srivastava4/publication/341114550_Concepts_of_Prakriti_Human_Constitution_and_its_Association_with_Hematological_Parameters_Body_Mass_Index_BMI_Blood_Groups_and_Genotypes/links/5eaf0d68a6fdcc7050a84f45/Concepts-of-Prakriti-Human-Constitution-and-its-Association-with-Hematological-Parameters-Body-Mass-Index-BMI-Blood-Groups-and-Genotypes.pdf https://www.researchgate.net/publication/308279992_Ayurgenomics_A_novel_approach_in_preventing_congenital_anomalies_A_review
Author Response
I include a new table
Table 1 (example of Prakriti and gene expression)
|
Prakriti Type |
Gene Expression |
Disease |
|
Kapha/Vata |
EGLN 1 higher |
High Altitude Pulmonary Edema |
|
Pitta |
EGLN 1 lower |
more adaptive for higher altitudes |
Reviewer 2 Report
Ayurgenomics is a new area of research that aims to provide a genomics framework to probe the principles of Ayurveda that are used for the preventive, predictive, personalised and participatory approach in their practice. Getting insights at the molecular level would enable its integration in the mainstream. These concepts are not readily acceptable to a Western audience and the author has really provided an unbiased commentary on this approach synthesizing evidence that are available in peer reviewed journals. The author has also provided a merit for this approach for integration in P4 medicine.
Author Response
no changes necessary
Reviewer 3 Report
Dear Robert,
In general, the review is well written. Couple of suggestions towards increasing the target audience with general interest in stratified medicine, are as follows:
- It may be good idea to make couple of graphical representation based figures which will convey the message with more clarity. For e.g. high altitude adaptation vis-a-vis presence of particular genotype.
- There can be a paragraph also touching upon the challenges towards scientific acceptibility of the concept and struggle of a researcher working in the trans-disciplinary domain. Many journals are still biased to exclude manuscripts for peer-review by finding the term Ayurveda in the title or abstract.
- Ayurveda-modern medicine bridging - is there a need for common terms as A. T, G, C irrespective of organism which we are studying. An opinion towards how that can be bridged and possible suggestions towards the same would be helpful.
Best wishes,
Author Response
I included a new table
Table 1 (example of Prakriti and gene expression)
|
Prakriti Type |
Gene Expression |
Disease |
|
Kapha/Vata |
EGLN 1 higher |
High Altitude Pulmonary Edema |
|
Pitta |
EGLN 1 lower |
more adaptive for higher altitudes |
Regarding bias towards Ayurveda I think I answered that in the paper. See below
Unfortunately, modern medicine has not recognized many of the useful preventative approaches of Ayurveda because of a cognitive bias against folk or traditional medicine. Even though traditional systems of medicine are still widely used in many countries around the world, more research on their preventative and therapeutic treatment programs is necessary [44].
Thank you for your suggestions